# Task-Conditional Adapter for Multi-Task Dense Prediction

Fengze Jiang
Zhejiang University
Hangzhou, China
22231079@zju.edu.cn

Shuling Wang
Zhejiang University
Hangzhou, China
11831041@zju.edu.cn

Xiaojin Gong*
Zhejiang University
Hangzhou, China
gongxj@zju.edu.cn

## ABSTRACT

Multi-task dense prediction plays an important role in the field of computer vision and has an abundant array of applications. Its main purpose is to reduce the amount of network training parameters by sharing network parameters while using the correlation between tasks to improve overall performance. We propose a task-conditional network that handles one task at a time and shares most network parameters to achieve these goals. Inspired by adapter tuning, we propose an adapter module that focuses on both spatial- and channel-wise information to extract features from the frozen encoder backbone. This approach not only reduces the number of training parameters, but also saves training time and memory resources by attaching a parallel adapter pathway to the encoder. We additionally use learnable task prompts to model different tasks and use these prompts to adjust some parameters of adapters to fit the network to diverse tasks. These task-conditional adapters are also applied to the decoder, which enables the entire network to switch between various tasks, producing better task-specific features and achieving excellent performance. Extensive experiments on two challenging multi-task benchmarks, NYUD-v2 and PASCAL-Context, show that our approach achieves state-of-the-art performance with excellent parameter, time, and memory efficiency. The code is available at https://github.com/jfzleo/Task-Conditional-Adapter.

## CCS CONCEPTS

• **Computing methodologies** → **Scene understanding**.

## KEYWORDS

Dense Prediction, Scene Understanding, Multi-task Learning, Transformer Adapters

**ACM Reference Format:**
Fengze Jiang, Shuling Wang, and Xiaojin Gong. 2024. Task-Conditional Adapter for Multi-Task Dense Prediction. In *Proceedings of the 32nd ACM International Conference on Multimedia (MM '24), October 28-November 1, 2024, Melbourne, VIC, Australia.* ACM, New York, NY, USA, 10 pages. https://doi.org/10.1145/3664647.3681581

---

*Xiaojin Gong is the corresponding author.

## 1 INTRODUCTION

Dense prediction is an important research direction in computer vision, involving tasks including semantic segmentation, depth estimation, edge detection, surface normal estimation, etc. These techniques are widely applied in various domains including autonomous driving [15, 23, 36], robotics [37, 51], virtual reality [31, 46, 52], among others. Different dense prediction tasks share part of the information when understanding the scene, which makes Multi-Task Dense Prediction (MTDP) a popular research direction. Typically, MTDP networks perform multiple prediction tasks in a unified framework. The advantages of this approach over training several single-task networks are twofold: first, learning a single network that can tackle multiple tasks will have fewer parameters, less computational cost, and less memory usage; second, complementary tasks will have mutual benefits [35, 38, 47].

With the rise of deep learning, researchers have recently proposed many deep learning-based MTDP methods. Typical MTDP methods handle multiple dense prediction tasks concurrently and exhibit excellent performance [5, 7, 13, 39, 42, 44–48, 53]. Nevertheless, they still require a substantial number of task-specific parameters to model distinct tasks, which hinders their ability to fully achieve the goal of parameter reduction [32, 39, 44, 46, 53]. An alternative approach to solve these problems is the task-conditional paradigm [20, 28–30, 35], which performs only one task at a time. These methods typically share the majority of their parameters across different tasks, greatly reducing the overall number of network parameters. This design endows them with better scalability when dealing with a multitude of tasks and makes the paradigm flexible for different application scenarios without extra computing resources. However, some of the methods concentrate solely on modulating the encoder [20, 30], while others focus only on adapting parts of the decoder [35]. This may not be sufficient for optimal performance in MTDP scenarios [29]. Recently, several task-conditional approaches [28, 29] have attempted to adjust both the encoder and decoder components simultaneously. These methods have shown improvements over previous task-conditional methods. Yet, there is still a performance gap between these methods and traditional MTDP methods [29, 35].

It is also worth noting that many recent works on parameter efficient transfer learning [4, 6, 18, 19] aim to adapt large powerful pre-trained networks to different downstream tasks, by inserting trainable adapters or prompts to a frozen transformer structure. These methods are notable for their efficiency, as they allow for the adaptation procedure with a small number of trainable parameters [17, 18]. This idea exactly coincides with our goal of reducing training parameters. Nevertheless, as stated in [6, 22], simply inserting prompts into a frozen backbone may be effective for image classification tasks, but it does not perform well for dense prediction tasks. What's more, adapters designed specifically for dense prediction

tasks [25, 49, 50] only focus on channel-dimensional adaptation, ignoring an important fact: channel and spatial information are both important for various dense prediction tasks [14, 47]. Additionally, traditional adapter tuning approaches involve inserting adapter modules sequentially into each transformer block[4, 19, 21]. While this does indeed reduce the number of parameters the network needs to train, the gradients still have to pass through the entire backbone during backpropagation [10, 49]. This results in these methods still consuming a significant amount of time and memory resources. In recent efforts [28, 29], the integration of adapters and prompts into the task-conditional paradigm has been explored. However, both of them only serve to guide the network's conditioning between tasks, and there is still a need to train all the parameters of the network, thus failing to realize the original purpose of adapters and prompts, which is to save training parameters.

In light of the aforementioned issues, our goal is to train an MTDP network with competitive performance, not only minimizing the number of training parameters but also reducing training time and memory consumption. To achieve this, our approach leverages the scalability advantages of the task-conditional paradigm. We propose a task-conditional adapter with two major functions. First, it extracts features from the frozen backbone network. This adapter differs from previous ones designed for dense prediction tasks by employing channel and spatial attention to focus on information in two dimensions. Specifically, we introduce channel attention and spatial attention modules from Convolutional Block Attention Module (CBAM) [41], which are responsible for focusing on channel features and spatial features respectively. These two modules are able to achieve competitive performance with only a small fraction of training parameters, enhancing their spatial localization accuracy and feature representation capability. Second, it modulates the whole network to accommodate various tasks. To represent different tasks, we assign a learnable task-specific prompt to each task type. Based on these prompts, we propose a simple yet effective task-conditional module for network parameter switching. This modulation strategy is applied to both the encoder and decoder, which enhances the model's flexibility and accuracy in handling multiple tasks. This enables the network to learn and adapt to the needs of diverse tasks more effectively, thereby achieving better performance in MTDP scenarios. Moreover, the proposed adapters are incorporated into the encoder in a parallel manner, forming a gradient highway [49]. By freezing the backbone network and avoiding the propagation of gradients through it during training, this approach not only conserves the count of parameters that need to be trained but also significantly cuts down on both the time and memory resources required for training. The adapter is connected in series behind each transformer block for the decoder, performing more direct modulation.

The application of the aforementioned approaches results in a parameter and computationally efficient task-conditional network. It achieves state-of-the-art performance in task-conditional methods with a similar number of training parameters, and compared to traditional MTDP methods, our proposed approach achieves a comparable level of performance with significantly lesser training parameters, time, and memory consumption.

In summary, the main contributions of this work are as follows:

- We propose a task-conditional MTDP network, which switches between various tasks based on task-specific prompts. Our model shares almost all of its parameters and reduces the task-specific parameters to a considerable degree.
- We design a novel transformer adapter that learns channel- and spatial-wise features at the same time. We freeze the backbone encoder and attach parallel adapter layers to it, reducing training parameters, training time, and memory usage further. The adapters also adjust part of their parameters according to learnable task prompts and are employed in the decoder as well to execute the task conditioning strategy on the entire model.
- Extensive experiments are done on two challenging multi-task dense prediction benchmarks, i.e., PASCAL-Context and NYUD-v2. The results show that our method achieves state-of-the-art performance and only requires training a small fraction of the network parameters.

## 2 RELATED WORK

### 2.1 Multi-task Learning for Dense Prediction

As a popular topic in computer vision, multi-task learning (MTL) aims to tackle multiple tasks while maintaining parameter efficiency and computational efficiency [3, 8, 38]. Many previous works [32, 39, 42, 53] have made numerous attempts in different aspects of this field. In particular, Cross-stitch networks [32] share information among the encoders of various single-task prediction networks, by incorporating activation layers from the networks. PAD-Net [42] uses a shared encoder among different tasks, and applies multi-modal distillation to perform cross-task interaction in the decoding phase. Taking it a step further, MTI-Net [39] utilizes a shared backbone that extracts multi-scale features, explicitly performing cross-task information interaction at multiple scales by employing a feature propagation module at each level.

The aforementioned methods are mostly implemented based on CNNs. With the promising performance of transformers [11, 26] in the domain of computer vision, it has been introduced in many recent researches[1, 5, 7, 13, 43–48]. Specifically, InvPT [46] leverages the long-range perceptual capabilities of Vision Transformer (ViT) structure by a carefully designed transformer decoder that models the spatial and cross-task relationships simultaneously. Taskprompter [47] proposes spatial and channel task prompts, which help the model to learn task-specific information and perform cross-task information interaction in both aspects. In addition, there are many excellent works that mainly focus on designing various network structures based on Mixture of Experts (MoE) [5, 7, 13, 45, 48], which is an alternative way for MTL. In particular, TaskExpert [48] utilizes Memorial MoE to equip the model with cross-layer interactions. MLoRE [45] proposes low-rank experts to enlarge the capacity of feature representations. Although these methods have achieved impressive performance, they suffer from large parameter numbers and high computational costs due to either employing different modules for different tasks or using different features to represent them. In this work, the network shares most of the parameters among tasks, and the backbone encoder is frozen during the training phase. The proposed adapter extracts spatial and channel information from the frozen backbone. Thus,

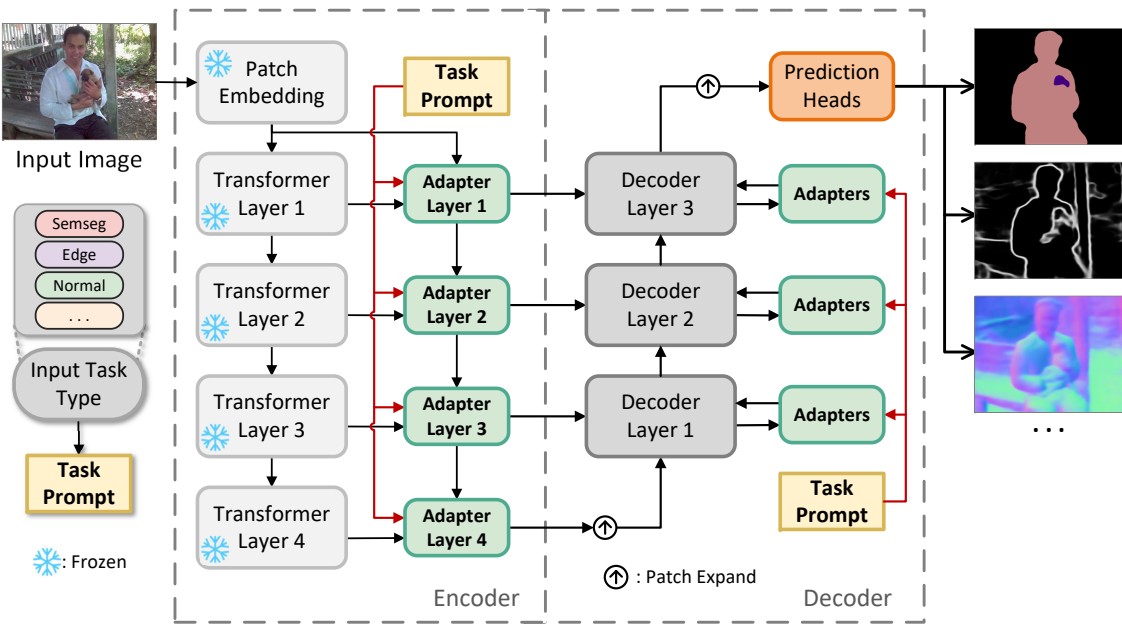

**Figure 1: Network overview. The proposed network is a single-encoder-single-decoder architecture. We utilize adapters to transfer the frozen backbone transformer encoder to dense prediction tasks and condition the whole network among tasks. The encoder is attached with parallel adapter layers, while the decoder has adapters sequentially inserted into it. Each task type is assigned with a trainable task-specific prompt. These prompts are used to condition the adapters in both the encoder and decoder, enabling the network to produce better task-specific features.**

we can save a significant number of training parameters and reduce computational costs while achieving excellent performance.

## 2.2 Task-conditional Paradigm

Many MTDP methods output predictions of all tasks by a single forward pass [32, 39, 42, 53]. Another type of MTDP method is based on task-conditional architectures [20, 30, 35], which execute only one task at a time. Typically, these methods involve employing different modules or adjusting network weights for different tasks. To be more specific, ASTMT [30] and RCM [20] utilize task-conditional encoders, which perform model adaptation by attention modules and model reparameterization, respectively. Instead of modifying the encoder, TSNs [35] conditions the task features during the decoding phase for different tasks, which allows the model to specialize to different tasks while still benefiting from the shared representation learned by the encoder. However, these methods either focus only on the encoder [20, 30] or only on the decoder [35], which limits their capacity to extract better task-specific features by modulating the entire network. Recently emerged task-conditional networks [28, 29] have been dedicated to adjusting both the encoder and decoder simultaneously, achieving better performance compared to the aforementioned approaches. Specifically, PGT [29] introduces task-specific prompts to model different tasks, and directly incorporate them in the self-attention mechanism to condition the whole network across multiple tasks. Nevertheless, visual prompts, as mentioned in [6, 22], are designed for image classification and perform suboptimally in dense prediction tasks because simply introducing prompts cannot fully model or represent all the fine-grained information required for dense prediction. TIT [28] also conditions

the encoder and decoder at the same time, with the guidance of task indicating matrix and vector, respectively. However, it lacks a unified representation for each task, as it employs different representational forms for the tasks in the encoder and decoder. In this paper, we achieve state-of-the-art performance by simultaneously conditioning the encoder and decoder based on the task type. We attain this by modeling diverse tasks using consistent, task-specific prompts and uniformly modulating task-conditional adapters in the encoder and decoder, improving the network's adaptability to various tasks while obtaining better task-specific features.

## 2.3 Adapters and Prompts for Vision Transformer

Adapters are a few trainable modules attached to large transformer models, and prompts are specific templates to reformulate downstream tasks. They both aim to transfer powerful pre-trained networks to different tasks while maintaining parameter efficiency. They are first introduced for language tasks [17, 24, 54], and many recent researches [1, 4, 6, 19, 25, 28, 49, 50] have shown that they also perform well in dense prediction tasks with vision transformers. Specifically, ViT-Adapter [6] injects features extracted from the feed-forward network into each transformer block for different dense prediction tasks such as object detection and semantic segmentation. Furthermore, Yin et al. [49] introduced a parallel adapter architecture that enables the model to adapt to diverse tasks effectively. This architecture directly extracts and adapts multi-scale features from the frozen hierarchical transformer backbone, further saving time and memory resources. Moreover, Bhattacharjee et al.

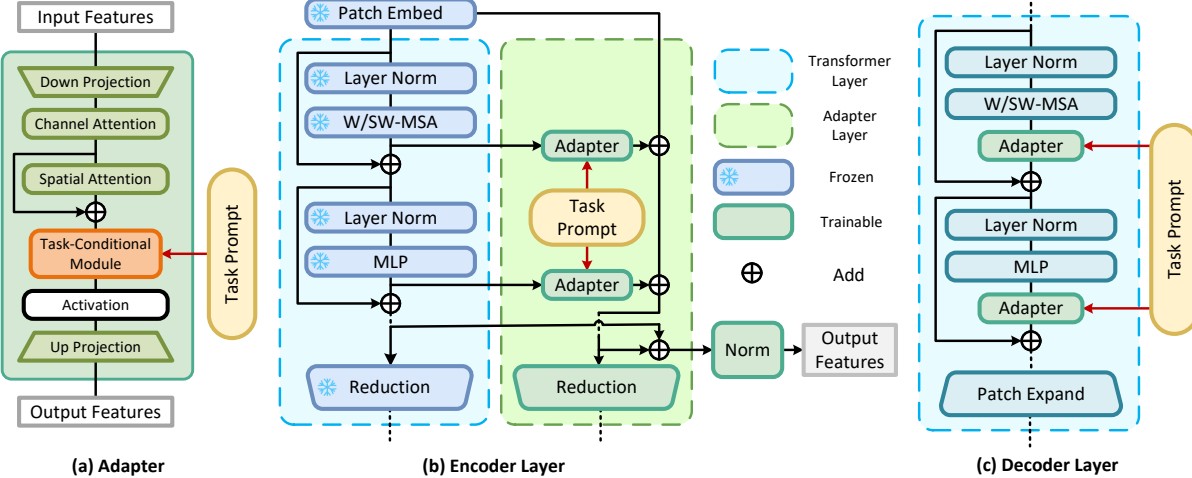

**Figure 2: The proposed adapter (a), encoder layer (b) and decoder layer (c). The proposed adapters (a) integrate channel and spatial attention modules to capture channel- and spatial-wise features. The task-conditional module is guided by a task prompt to perform task-specific conditioning. In (b), blue rectangles represent frozen backbone transformer modules, and green rectangles denote the trainable adapters. The adapters form a parallel pathway to extract features from the frozen backbone efficiently. It not only reduces the number of training parameters but also saves training time and memory consumption. In (c), adapters are inserted in the decoder layers sequentially, which performs direct modulation on the decoder.**

[1] proposed an adapter bottleneck that utilizes a task-adapted attention mechanism to perform cross-task reasoning, which enables the model to tackle multiple dense prediction tasks at the same time. Nevertheless, these approaches only focus on modulating channel dimension. For MTDP tasks, spatial dimension information is as crucial as channel dimension information [14, 47]. Performing channel-dimensional modulation alone overlooks spatial feature interactions, leading to a suboptimal performance. To address this, we introduce channel and spatial modules as the adapter kernel which pays attention to channel- and spatial-dimensional information respectively. This dual attention mechanism enhances the feature extraction capabilities across both dimensions, thereby improving the perception and modeling of the environment. In addition to this, we adopt a parallel adapter architecture to further reduce both training time and memory consumption.

## 3 METHOD

In this section, we first overview the network architecture we propose. Second, we detail the adapter module, which integrates the channel and spatial attention module, as well as the task-conditional module. Next, we introduce the encoder and decoder equipped with the proposed adapter. Finally, we give a brief introduction to task-specific heads and training loss.

### 3.1 Overall Architecture

As shown in Figure 1, the network constitutes a multi-scale single-encoder-single-decoder architecture. To overcome the parameter bloating problem for traditional multi-task structures [29, 35], we adopt a task-conditional network paradigm and training approach [20, 30, 35], which takes a single RGB image and the specified task type as input, and outputs the corresponding task prediction map. By sharing the vast majority of its parameters across different tasks,

the network enhances parameter efficiency and reduces the overall number of trainable parameters.

As laid out in the introduction, we put forth that modulating both the encoder and the decoder is necessary. Therefore, we attach adapter layers to the encoder in a parallel manner and insert adapter modules into the decoder sequentially. All of the adapters are modulated among tasks, aiming at enhancing the adaptability of the entire network to diverse tasks. Detailed explanations of these modules will be provided below. To explicitly model the tasks, we assign a learnable task-specific prompt $e_t \in \mathbb{R}^d \times 1$ to each task $t$. The input of our network would then be a pair of image and task prompt, i.e., $(X, e_t)$. The image $X$ is then fed into the pre-trained backbone to produce the required features. The task prompt $e_t$ is used to guide some parameters of the adapters to switch when conducting different tasks.

### 3.2 Task-Conditional Adapter

Many recent works on adapters have shown their potential in various vision tasks with remarkable efficiency [4, 25, 49, 50]. They attach adapters to specific locations of the pre-trained backbone model, and only train adapters and layer-normalizations, while the remaining parameters remain fixed. Formally, typical adapter layer $A$ consists of a down-projection $D \in \mathbb{R}^{k \times r}$, a GeLU non-linearity [16], and an up-projection $U \in \mathbb{R}^{r \times k}$, where $k$ is the input dimension, and $r$ is the kernel dimension for the adapter layer. Thus, typical adapter layers can be written as

$$A(x) = U(GeLU(D(x))) + x, \tag{1}$$

where $x$ denotes the input feature.

Our task-conditional adapter serves two major functions: first, applying transfer learning techniques to the backbone network to tailor it for dense prediction tasks, and second, modulating the

| Model | Backbone | Param* M | Semseg mIoU ↑ | Depth RMSE ↓ | Normal mErr ↓ | Edge odsF ↑ |
|---|---|---|---|---|---|---|
| Cross-Stitch [32] | ResNet-50 | 80.1 | 44.22 | 0.5703 | - | - |
| PAD-Net [42] | ResNet-50 | 52.6 | 50.23 | 0.5818 | - | - |
| MTI-Net [39] | HRNet-18 | 27.2 | 38.61 | 0.5935 | - | - |
| InvPT [46] | ViT-L | 402.1 | 53.56 | 0.5183 | 19.04 | 78.10 |
| TaskPrompter [47] | ViT-L | 492.2 | 55.30 | 0.5152 | 18.47 | 78.20 |
| TaskExpert [48] | ViT-L | - | 55.35 | 0.5157 | 18.54 | 78.40 |
| MLoRE [45] | ViT-L | - | 55.96 | 0.5076 | 18.33 | 78.43 |
| ASTMT [30] | ResNet-50 | 45.0 | 32.16 | 0.5700 | 23.18 | 74.50 |
| RCM [20] | ResNet-18 | 39.0 | 34.20 | 0.5700 | 22.41 | 68.44 |
| TSNs [35] | Swin-T | 39.2 | 32.38 | 0.6874 | 22.25 | 75.69 |
| TIT [28] | Swin-T | 30.9 | 41.36 | 0.5925 | 19.68 | 77.30 |
| PGT [29] | Swin-T | 28.4 | 41.61 | 0.5900 | 20.06 | 77.05 |
| PGT [29] | Swin-B | - | 47.42 | 0.5502 | 19.12 | **78.28** |
| Ours | Swin-T | **12.4** | 46.08 | 0.5902 | 19.66 | 77.60 |
| Ours | Swin-B | 17.6 | 53.30 | 0.5235 | 19.07 | 77.90 |
| Ours | Swin-L | 38.4 | **54.56** | **0.5197** | **18.65** | 78.00 |

| Model | Backbone | Param* M | Semseg mIoU ↑ | Parts mIoU ↑ | Sal maxF ↑ | Normal mErr ↓ | Edge odsF ↑ |
|---|---|---|---|---|---|---|---|
| Cross-Stitch [32] | ResNet-18 | 80.3 | 66.12 | 60.66 | 66.81 | 13.89 | 69.90 |
| PAD-Net [42] | ResNet-18 | 32.1 | 63.23 | 59.34 | 64.31 | 15.20 | 60.20 |
| MTI-Net [39] | HRNet-18 | 15.7 | 64.35 | 62.10 | 68.02 | 14.78 | 73.40 |
| InvPT [46] | ViT-L | - | 79.03 | 67.61 | 84.81 | 14.15 | 73.00 |
| Taskprompter [47] | ViT-L | 493.0 | 80.89 | 68.89 | 84.83 | 13.72 | 73.50 |
| TaskExpert [48] | ViT-L | 420 | 80.64 | 69.42 | 84.87 | 13.56 | 73.30 |
| MLoRE [45] | ViT-L | 407 | 81.41 | 70.52 | 84.90 | 13.51 | 75.42 |
| ASTMT [30] | ResNet-50 | 49.4 | 68.00 | 61.12 | 65.71 | 14.68 | 72.40 |
| RCM [20] | ResNet-18 | 46.1 | 65.70 | 58.12 | 66.38 | **13.70** | 71.30 |
| TSNs [35] | Swin-T | 39.1 | 67.30 | 61.11 | 64.29 | 14.55 | **74.04** |
| TIT [28] | Swin-T | 31.3 | 70.04 | 62.68 | 66.14 | 14.43 | 73.91 |
| PGT [29] | Swin-T | 28.5 | 67.58 | 62.58 | 65.59 | 13.95 | 73.93 |
| M³ViT [13] | ViT-S | 42 | 72.80 | 62.10 | - | 14.50 | 71.70 |
| Mod-Squad [7] | ViT-S | 50 | 74.10 | 62.70 | - | **13.70** | 72.00 |
| Ours | Swin-T | **12.7** | 77.36 | 65.32 | **84.05** | 14.18 | 73.00 |
| Ours | Swin-L | 38.7 | **82.08** | **69.85** | 84.06 | 13.73 | 73.30 |

(a) Comparison with state-of-the-arts on NYUD-v2      (b) Comparison with state-of-the-arts on PASCAL-Context

**Table 1: Comparison with state-of-the-art methods on NYUD-v2 (a) and PASCAL-Context (b). '↑': lower better, '↓': higher better, '∗': number of training parameters. The upper parts are traditional MTDP methods and the lower parts are task-conditional methods. Our method clearly outperforms all of the task-conditional methods by a large margin, while achieving competitive results among the traditional methods.**

backbone's output features specifically for each distinct task. So as shown in Figure 2 (a), the adapter structure primarily consists of two parts: the adaptation module and the task-conditional module.

For the adaptation part, our approach differs from standard adapter architectures in two main aspects. First, we introduce a kernel module between down-projection and GeLU activation, following [21, 25, 49]. Second, as previously discussed, both spatial and channel information is indispensable for a wide range of dense prediction tasks [14, 47]. So we adopt the kernel structure from Convolutional Block Attention Module (CBAM) [41]. The design of this kernel pays attention to both the channel dimension and spatial dimension information of features.

To be specific, we borrow the channel and spatial attention module from CBAM [41] to focus on information about channel and spatial dimension, respectively. They use average pooling and max pooling to aggregate this information to obtain channel and spatial weight matrices, and finally diffuse the channel and spatial attention information into the input features through element-wise multiplication. Formally, given the down-projected feature $y \in \mathbb{R}^{r_l \times H_l \times W_l}$, we define channel attention map as $M_c \in \mathbb{R}^{r_l \times 1 \times 1}$ and spatial attention map as $M_s \in \mathbb{R}^{r_l \times H_l \times W_l}$, where $r_l$ is the kernel dimension at adapter layer $l$, $H_l$ and $W_l$ is the height and width of the input feature map, respectively. Thus, the spatial-channel kernel can be written as

$$
\begin{aligned}
y' &= M_c(y) \otimes y, \\
y'' &= M_s(y') \otimes y' + y',
\end{aligned}
\quad (2)
$$

where $\otimes$ denotes the Hadamard product with broadcasting. Moreover, channel attention map $M_c$ can be obtained by

$$M_c(y) = \sigma(MLP(AvgPool_c(y)) + MLP(MaxPool_c(y))), \quad (3)$$

and spatial attention map $M_s$ can be obtained by

$$M_s(y) = \sigma\left(f^{7 \times 7}([AvgPool_s(y); MaxPool_s(y)])\right), \quad (4)$$

where $\sigma(\cdot)$ denotes sigmoid activation, $f^{7 \times 7}$ denotes convolutional layer with a filter size of $7 \times 7$, [; ] denotes concatenation operation,

$c$ and $s$ denotes the pooling layer for channel and spatial dimension, respectively. Thus, leveraging spatial and channel attention modules, the adapters are able to focus on both spatial and channel dimensions. This dual focus allows the adapter to better capture channel and spatial information, leading to better performance in various dense prediction tasks.

For the task-conditional part, we propose a simple yet effective task-conditional module that enables the adapter to switch between different tasks, which enables the entire network to adapt to a variety of dense prediction tasks. We achieve this by adapting the features of the entire kernel part based on the additional task-specific prompt input. Specifically, given the input task prompt $e_t \in \mathbb{R}^{d \times 1}$ which represents task $t$, this module uses two linear layers, $W_\gamma \in \mathbb{R}^{r_l \times d}$ and $W_\beta \in \mathbb{R}^{r_l \times d}$, to learn two normalization weight vectors to modulate the output of the spatial-channel kernel $y'' \in \mathbb{R}^{r_l \times H_l W_l}$. Therefore, the task-conditional module $TCM$ can be written as

$$TCM(y'', e_t) = W_\gamma e_t \otimes y'' + W_\beta e_t. \quad (5)$$

This enables the adapters to condition their parameters between tasks, thereby enabling the network to extract superior task-specific features, which in turn enhances the network's comprehensive understanding of the environment. Then the whole adapter module can be written as

$$A(y, e_t) = U(GeLU(TCM(K(D(y)), e_t)) + y, \quad (6)$$

where $K$ denotes the spatial-channel kernel introduced above. Moreover, as illustrated in Figure 1, by applying the proposed adapter to both the encoder and decoder, the network is able to adapt efficiently and flexibly to different task requirements. This flexibility is crucial for multi-task learning scenarios where the network must perform well across a variety of tasks with varying demands.

### 3.3 Task-Adapted Encoder

Our encoder is built upon Swin Transformer [26]. As discussed above, we freeze the backbone encoder and utilize adapter modules to further reduce the training parameters, and serve as a modulator

| EA | DA | TCM | CA | SA | Param* M | Semseg mIoU↑ | Depth RMSE↓ | Normal mErr↓ | Edge odsF↑ | MTL Gain $\Delta_m$ [%] ↑ |
|----|----|-----|----|----|----------|--------------|-------------|--------------|------------|---------------------------|
| | | STL Baseline | | | 351.9 | 52.19 | 0.5433 | 20.04 | 78.00 | 0.00 |
| | | MTL Baseline | | | 88.8 | 51.61 | 0.5512 | 20.47 | 76.50 | -1.66 |
| ✓ | | | | | 17.5 | 49.45 | 0.5550 | 20.55 | 77.20 | -2.74 |
| ✓ | | ✓ | | | 17.5 | 51.38 | 0.5379 | 20.00 | 77.50 | -0.25 |
| | ✓ | ✓ | | | 17.6 | 50.53 | 0.5606 | 20.47 | 77.40 | -2.31 |
| ✓ | ✓ | ✓ | | | 17.6 | 52.00 | 0.5439 | 19.97 | 77.50 | -0.19 |
| ✓ | ✓ | ✓ | ✓ | | 17.6 | 52.07 | 0.5326 | 19.67 | 77.50 | 0.74 |
| ✓ | ✓ | ✓ | ✓ | ✓ | 17.6 | **53.30** | **0.5235** | **19.07** | **77.90** | **2.82** |

**Table 2: Effectiveness of different modules on NYUD-v2. "EA" means encoder adapter, "TCM" means task-conditional module, "CA" means channel attention module, "SA" means spatial attention module and "DA" means decoder adapter.**

that adjusts the encoder to adapt to various tasks. Different from previous works on adapters [4, 17, 25, 50] which insert adapter modules directly into backbone transformer blocks, our encoder structure adopts a parallel adapter architecture [49] as illustrated in Figure 2(b). The blue boxes denote the frozen modules in the backbone transformer, while the green boxes are the proposed adapters that are trainable.

The proposed task-conditional adapter extracts features from each attention layer and each MLP layer in the backbone, and adjusts the features according to task-specific prompt $e_t$, utilizing the task-conditional module mentioned above. The extracted and adapted features would then be added into a parallel pathway. Before the reduction module of each layer, the features of the adapter pathway are summed with the backbone features to obtain the encoder output features, which follows the residual structure of standard adapter methods and imitates skip connections in transformer blocks [11, 26]. This parallel adapter pathway structure serves as a gradient backpropagation highway [49]. In this way, only the gradients on this highway need to be computed, which allows the encoder to avoid calculating the gradients of frozen parameters, which not only saves the number of training parameters but also reduces memory usage and training time.

The output features then pass through independent normalization layers specific to each scale, thereby producing the final multi-scale feature representations. Subsequently, these refined features serve as the input to the task-conditioned decoder, whose details will be unfolded below.

## 3.4 Task-Conditioned Decoder

Following the idea in [1, 2, 26], our decoder consists of four stages, with a patch expand module [2] between each stage to double the spatial resolution and halve the channel dimension. The last three stages of the decoder each contain two transformer blocks. Different from the encoder backbone, our decoder does not have a pre-trained model, so fine-tuning is required. Consequently, there is no need for a parallel pathway in the decoder to shorten the gradient backpropagation path. Moreover, using a parallel structure in a decoder that requires fine-tuning would weaken the modulating effect of the adapter on the decoder, leading to suboptimal information reconstruction and decoding outcomes.

Therefore, as illustrated in Figure 2(c), we follow typical adapter structures [4, 21, 25], sequentially connecting the adapters directly

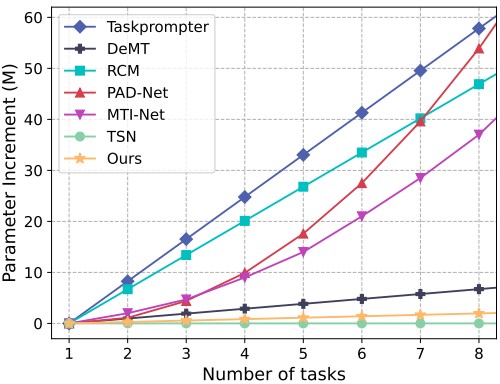

**Figure 3: Parameter efficiency analysis. Our method, while not entirely independent of the number of tasks, is minimally affected by it.**

after each attention module and MLP module in the decoder. This method applies direct modulation to each layer of the decoder, enabling more fine-grained adjustments to the decoder. This allows it to generate more representative features based on task-specific information, thereby improving performance.

After expanding the final output to the original image size $H \times W$, we apply an independent convolutional layer specifically to each task to obtain the final prediction map with a shape of $K \times H \times W$, where $K$ denotes the output channels for different tasks.

## 3.5 Training Loss

We perform unified end-to-end training on adapters and the decoder. Each training step is only performed on one task, and only one ground truth map is utilized to calculate losses. For a fair comparison, we define our loss function following previous works [44, 46, 47]. We adopt task-specific loss $\alpha_t \mathcal{L}_t$ with weight $\alpha_t$ for task $t$, and align them with [46, 47].

## 4 EXPERIMENTS

## 4.1 Experimental Setup

**Datasets**. We evaluate the proposed network on two widely used datasets on multi-task dense prediction, i.e., NYUD-v2 [34] and PASCAL-Context [33]. **NYUD-v2** dataset consists of 795 training and 654 testing images of various indoor scenes such as offices

and living rooms. It provides labels for four tasks of edge detection (*Edge*), semantic segmentation (*SemSeg*), surface normal estimation (*Normal*) and depth estimation (*Depth*). **PASCAL-Context** dataset is established from PASCAL dataset [12]. It contains 4,998 images in the training split and 5,105 in the testing split. The content of these images includes both indoor and outdoor scenes. This dataset provides pixel-wise annotations for semantic segmentation, edge detection, surface normal estimation, human parts segmentation (*Parts*) and saliency detection (*Sal*).

**Evaluation metrics**. We adopt widely used evaluation metrics following existing works [40, 46, 47]. Specifically, semantic segmentation and human parts segmentation utilize mean Intersection over Union (*mIoU*) metric to evaluate the predictive performance. Monocular depth estimation uses Root Mean Square Error (*RMSE*) for evaluation. Surface normal estimation uses mean error (*mErr*) of predicted angles. Saliency detection is evaluated with maximal F-measure (*maxF*). Edge detection adopts optimal-dataset-scale F-measure (*osdF*). Moreover, we adopt $\Delta_m$ evaluation metric introduced in [30] to measure multi-task gain (*MTL Gain*).

**Implementation details**. Our method is built upon Swin Transformer [26] backbones pre-trained on ImageNet-22K [9]. We use two SwinBlocks at each decoding level, set task-specific prompt length $N_e$ to 128, and down projection ratio $\rho$ to 16 unless stated otherwise. To preserve robustness within the adapter layer, we incorporate the sharing of spatial attention modules and task-conditional modules internally within each adapter layer. We train our model for 100 epochs on PASCAL-Context dataset and 500 epochs on NYUD-v2 dataset. Our experiments were conducted on 4 NVIDIA RTX A6000 GPUs with a batch size of 4. We use AdamW optimizer [27] and set the learning rate to $1 \times 10^{-4}$.

## 4.2 Parameter Efficiency Analysis

As one of the goals of this work, parameter efficiency needs to be carefully analyzed. According to previous studies [29, 35], the total number of parameters within a network directly increases with the addition of more task-specific parameters. Thus, in terms of handling numerous tasks, it becomes crucial to limit the number of task-specific modules and the parameters they contain.

From Figure 3, it can be observed that some traditional MTDP methods, such as PAD-Net [53] and MTI-Net [39], have a parameter count that scales quadratically with the number of tasks. This is because these methods design independent modules to model the relationships between each pair of tasks. Other approaches, like Taskprompter [47], DeMT [44], ASTMT [30], and RCM [20], all incorporate task-specific modules to model different tasks, resulting in a linear increase in the number of parameters as the number of tasks grows. It is noteworthy that TSNs [35] is an excellent task-conditional architecture where all parameters are shared across tasks, hence its parameter count remains unchanged regardless of the number of tasks. In contrast, our method includes task-specific parameters that consist only of learnable task prompts and simple task-specific output heads. Therefore, the incremental increase in parameters compared to the overall network parameters is negligible. Based on the analysis of parameter efficiency, we can demonstrate that our proposed network shares the vast majority of parameters across different tasks, with the total parameter count

| Length | Param* M | Semseg mIoU↑ | Depth RMSE↓ | Normal mErr↓ | Edge odsF↑ |
|---|---|---|---|---|---|
| 32 | **17.6** | 52.79 | 0.5364 | **19.03** | **77.90** |
| 64 | **17.6** | 52.86 | 0.5357 | 19.10 | 77.80 |
| 128 | 17.7 | **53.30** | **0.5235** | 19.07 | **77.90** |
| 256 | 17.7 | 52.20 | 0.5285 | 19.06 | 77.80 |

**Table 3: Ablations for task prompt length on NYUD-v2.**

| Ratio $\rho$ | Param* M | Semseg mIoU↑ | Depth RMSE↓ | Normal mErr↓ | Edge odsF↑ |
|---|---|---|---|---|---|
| 2 | 33.8 | 53.35 | **0.5197** | **18.66** | **78.10** |
| 4 | 23.9 | **54.02** | 0.5352 | 18.74 | 77.90 |
| 8 | 19.6 | 52.98 | 0.5265 | 18.95 | 77.70 |
| 16 | 17.6 | 53.30 | 0.5235 | 19.07 | 77.90 |
| 32 | **16.7** | 53.83 | 0.5241 | 19.37 | 77.90 |

**Table 4: Ablations for down-projection ratio on NYUD-v2.**

| Manner | Backbone | Param* M↓ | FLOPs G↓ | Time h↓ | Memory GiB↓ |
|---|---|---|---|---|---|
| InvPT [46] | ViT-L | 423 | 669 | 24.19 | 7.08 |
| Taskprompter [47] | ViT-L | 401 | 497 | 36.34 | 9.26 |
| ASTMT [30] | CNN | 365 | 501 | - | - |
| Sequential | Swin-L | **38.7** | **336** | 10.89 | 6.35 |
| Parallel | Swin-L | **38.7** | **336** | **9.74** | **4.17** |

**Table 5: Parameters, time, and memory.**

being minimally affected by an increase in the number of tasks, achieving remarkable success in saving network parameters.

## 4.3 Comparison With State-of-the-Art

In this section, we will compare the proposed method with state-of-the-art task-conditional methods, which share the same multi-task learning paradigm as our method. As a reference, we will also compare recent traditional MTDP approaches. The experimental results on NYUD-v2 and PASCAL-Context are reported in Table 1(a) and Table 1(b), respectively. Our model not only outperforms the current best task-conditional method, PGT [29], by a substantial 5% margin but also achieves it with only half the number of training parameters. Similarly, compared to the best traditional multi-task methods, our approach manages to match or even exceed their performance levels with a training parameter count that is only less than 10% of what those methods require. This strongly demonstrates the adaptability of our method to different dense prediction tasks and the outstanding effectiveness of our method in saving the number of parameters.

## 4.4 Ablation Studies

**Proposed modules**. We verify the effectiveness of our proposed modules on the NYUD-v2 dataset and report the results in Table 2. Based on Swin Transformer, we have built a strong baseline whose performance is comparable to existing MTDP models. Among them, "STL Baseline" is used to train a set of separate single-task models, and each model is only trained for one task. "MTL Baseline" shares encoder and decoder among tasks, and only uses task-specific prediction heads to deal with different tasks. Both of

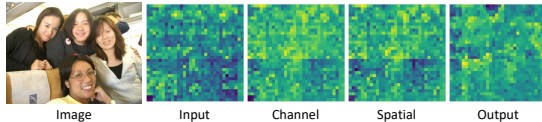

**Figure 4: Feature map visualization.**

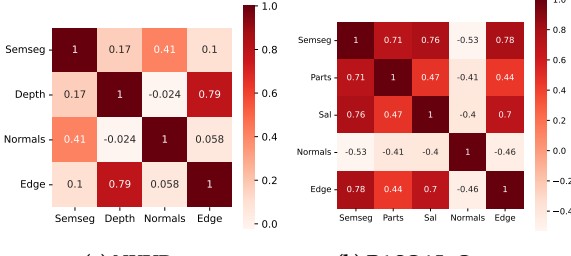

(a) NYUDv2  (b) PASCAL-Context

**Figure 5: Task prompt correlations.**

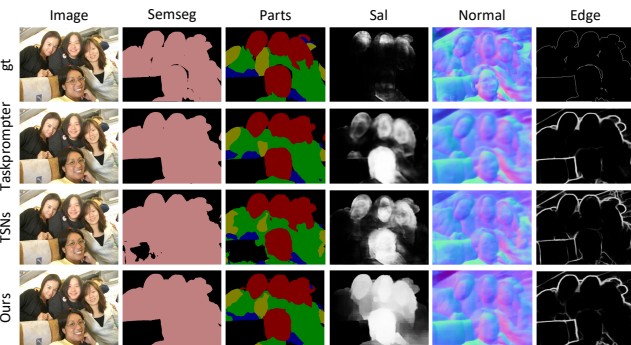

**Figure 6: Qualitative results.**

these two baseline models fine-tune all of their parameters. Inserting the task-conditional module into the adapters endows the entire module with the ability to adapt to different tasks, allowing the module to switch when processing different tasks. The introduction of channel and spatial attention further enhances the adapter's perception of these two dimensions of information, thereby further improving performance. The performance gap between adding adapters only to the decoder and adding them to both the encoder and decoder also validates our previous viewpoint that both the encoder and decoder need to be modulated for different tasks. For a clearer demonstration, Figure 4 shows how the proposed modules in the adapter enhance the feature maps. These experimental results strongly demonstrate the effectiveness of the various modules proposed.

**Task prompts**. The learnable task-specific prompts model the information of various tasks and are utilized to represent the corresponding tasks within each adapter. They play a vital role in guiding the modulation of the entire network. The length of the task prompts is directly related to the amount of information they can capture, necessitating a thorough analysis of their optimal length. Experimental results on the NYUD-v2 dataset, as presented in Table 3, indicate that the model's performance gradually improves with the increase in the length of the task prompts. This improvement reaches a peak when the length is set to 128, and beyond this point, further extending the length does not yield significant enhancements in model performance. An excessively large dimension may cause the task prompts to start modeling similar or repeated information within the tasks, leading to the problem of information redundancy. Increasing the length of the task prompts will also further increase the overall number of parameters in the network. Therefore, we set the task prompt length to 128. In addition, Figure 5 visualizes the cosine similarity of different task prompts, which provides concrete insights into task relationships and demonstrates that our learnable task prompts can effectively model different tasks.

**Down-projection ratio**. For adapters, the ratio of input dims to middle dims (the down-projection ratio $\rho$) is an important metric for balancing the number of parameters and model performance. The larger this ratio, the higher the parameter efficiency of the

adapter, but generally, the worse the model performance. Therefore, choosing the right value for this ratio is a question that is well worth researching. The results of our ablation study on $\rho$ are shown in Table 4. We can observe that the correlation between our model's parameter count and performance with respect to this ratio essentially aligns with the trend we just discussed. After consideration and trade-offs, we set the ratio to 16.

**Training time and memory consumption**. As mentioned above, the parallel connection manner of adapters helps to reduce training time and memory consumption. Therefore, we conduct ablation experiments on these two methods. Using the Swin-L setting and setting the batch size to 2, we train the network for 40,000 steps and record its time and memory consumption. The experimental results are shown in Table 5, and as a reference, we also conduct experiments on existing methods. Our proposed method outperforms other methods in terms of parameter count, memory footprint, and training time by a large margin. The parallel attachment of adapters to the encoder results in a 34% reduction in memory consumption and an 11% reduction in training time, compared to the sequential manner.

## 4.5 Qualitative Results

For an intuitive comparison, we present an analysis of our proposed method's performance by conducting a visual comparison on the PASCAL-Context dataset. We benchmark our approach against two categories of existing methods: traditional MTDP methods, represented by Taskprompter [47], and task-conditional methods, exemplified by TSNs [35]. As shown in Figure 6, it is clear that our proposed method presents more reasonable and accurate results in semantic segmentation and human parts segmentation tasks compared to the existing methods, and it also shows good performance in other tasks.

## 5 CONCLUSION

In this paper, we introduce a task-conditional adapter designed to extract features from a frozen backbone network and condition the entire network based on a trainable task prompt. These adapters are connected in parallel to the encoder, which not only saves on training parameters but also reduces training time and memory footprint. For the decoder, adapters are inserted sequentially to perform more direct modulation. Our experiments demonstrate the effectiveness of the individual components we propose and showcase the parameter, time, and memory efficiency of our approach.

# ACKNOWLEDGMENTS

This research was funded by Zhejiang Province Pioneer Research and Development Project "Research on Multi-modal Traffic Accident Holographic Restoration and Scene Database Construction Based on Vehicle-cloud Intersection" (Grant No. 2024C01017).

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
