# OpenReview forum: "Task-Conditional Adapter for Multi-Task Dense Prediction"
_acmmm.org/ACMMM/2024/Conference — MM2024 Poster_

### Official Review · Reviewer_uzJ2 · 2024-05-01

**Rating:** 5
**Confidence:** 4

**Summary:**

This paper proposes a task-conditional model for multi-task dense prediction, introducing an adapter module to extract spatial- and channel-wise information from the pretrained encoder backbone. The encoder employs a parallel pathway to reduce training time and memory overhead. Learnable task prompts are used as a conditioning strategy on the adapters. Experimental results on two public datasets NYUD-v2 and PASCAL-Context validate the effectiveness of the proposed approach.

**Strengths:**

1. The paper's presentation is clear, and its framework is straightforward.
2. The motivation behind the paper is sound.  While many recent works in leverage transformers to improve MTL performance, they often come with drawbacks such as increased model parameters, GPU memory usage, and training time. This paper proposes a refreshing solution by fine-tuning only the adapter modules and decoders.
3. The experiments demonstrate that the proposed model outperforms previous SOTA methods TIT and PGT by a clear margin, while maintaining parameter efficiency. Abundant ablation studies also provide insights into model design.

**Limitations:**

1. Including visualizations of feature maps and task prompts would strengthen the paper by validating the effectiveness of the proposed adapter module in extracting channel and spatial features, and the learnable task prompts in modeling task relationships. The authors may refer to TaskPrompter and PGT paper for examples of visualizations.
2. While Table 1 compares the number of training parameters across various methods, it would be beneficial to also include the number of all parameters and computation cost (FLOPs), as they are critical factors in evaluating MTDP methods, as pointed out by previous works.
3. The authors are recommended to provide the MTL Gain metric in Table 1 to offer a clearer illustration of the overall MTL performance. For instance, by using the same backbone Swin-T, the authors could compare the MTL Gains of TSNs, TIT, PGT, and proposed model.
4. More hyperparameter specifications such as learning rate and optimizer could be included in Implementation details section. The authors are also encouraged to open source code to enhance reproducibility.
5. Some important references are missing, particularly some MTDP works based on Mixture-of-Experts, which also handle one task at a time similar to task-conditional methods.
[1] M³ViT: Mixture-of-Experts Vision Transformer for Efficient Multi-task Learning with Model-Accelerator Co-design
[2] AdaMV-MoE: Adaptive Multi-Task Vision Mixture-of-Experts
[3] Mod-Squad: Designing Mixtures of Experts As Modular Multi-Task Learners
[4] TaskExpert: Dynamically Assembling Multi-Task Representations with Memorial Mixture-of-Experts

**Suitability:**

3

---

### Official Review · Reviewer_H1WX · 2024-05-25

**Rating:** 3
**Confidence:** 3

**Summary:**

This paper utilizes adapters to transfer the frozen backbone transformer encoder to dense prediction tasks and condition the whole network among tasks. The encoder is attached with parallel adapter layers, while the decoder has adapters sequentially inserted into it. Each task
type is assigned with a trainable task-specific prompt. These prompts are used to condition the adapters in both the encoder and decoder, enabling the network to produce better task-specific features.

**Strengths:**

The proposed method can train an MTDP network with competitive performance, not only minimizing the number of training parameters but also reducing training time and memory consumption.

**Limitations:**

1. This work looks like an assembly of existing methods. For example, 1) equipping the adapter with channel attention and spatial attention modules (Convolutional Block Attention Modules (CBAM)), which can be hard to consider as a novelty; 2) TaskPrompter, Spatial-Channel  information, adapter all have been employed in MTL; 3) Why does this paper sometimes use a parallel adaptor and sometimes serial.

2. More SOTA MTL methods are not discussed and compared, such as [1][2], whose performance are much higher than TaskPrompter.

3. To show the advantages of Param/Time/Memory of the proposed method, The authors are suggested to compare it with other methods.

[1] Adamv-moe: Adaptive multi-task vision mixture-of-experts
[2] Multi-Task Dense Prediction via Mixture of Low-Rank Experts

**Suitability:**

3

---

### Official Review · Reviewer_uHHV · 2024-05-25

**Rating:** 4
**Confidence:** 4

**Summary:**

This paper proposes a novel framework for multi-task dense prediction tasks, which significantly reduces the number of trainable parameters by using task-specific prompts and adapters. It introduces channel and spatial attention mechanisms in both encoder and decoder adapters, enhancing task-specific feature extraction across various tasks such as semantic segmentation and depth estimation. In general, this is an interesting paper.

**Strengths:**

1. The proposed method proposes innovative integration of adapters: utilizes a combination of channel and spatial attention mechanisms tailored to the multi-task learning, enhancing performance while maintaining parameter efficiency.
2. The proposed method is efficient in terms of parameter usage. It achieves substantial reductions in the number of trainable parameters compared to traditional methods, without compromising on the model's performance.
3. Extensive experiments were conducted on two standard datasets. Ablation studies are also provided to show the effectiveness of the key designs of the method.

**Limitations:**

1. I have a question about the decoder: do the adapters share parameters?
2. In lines 636-637, it is mentioned that "We adopt task-specific loss.. and align them with [41, 42]". In my opinion, this is not very clear and more details should be provided. Since some methods in Table 1 do not use these weighted losses, it is questionable whether the comparisons are fair. It might help if ablation studies on losses are provided to give a idea of what role the losses are playing.
3. It will be better if the authors could provide the visualization results to show the role of the two attentions?

Tiny things: 1. There is no punctuation in Eq.1; 2. The two sub-tables in Table 1 have different heights. It will look better if both sub-tables can be aligned.

**Suitability:**

3

---

### Meta-Review · Area_Chair_Mjai · 2024-07-03

**Recommendation:** Accept (Poster)
**Confidence:** 5

**Metareview:**

This submission presents a task-conditional network for multi-task dense prediction, which reduces network training parameters and improves performance by sharing network parameters and leveraging task correlations. Inspired by adapter tuning, it introduces an adapter module for efficient feature extraction and learnable task prompts, achieving good performance on NYUD-v2 and PASCAL-Context benchmarks with significant parameter, time, and memory efficiency.

All reviewers acknowledge that the proposed method achieves good performance without increasing network parameters or adding inference cost. However, Reviewer H1WX raises a concern about the novelty of the proposed method, and Reviewers uHHV and uzJ2 ask for clarification on the experimental results. After the rebuttal, all reviewers agree on acceptance. Please ensure the revised version incorporates all suggested changes.